# Dissociating Autoantibody Responses against Ro52 Antigen in Patients with Anti-Synthetase or Anti-MDA5 Antibodies

**DOI:** 10.3390/diagnostics13243621

**Published:** 2023-12-08

**Authors:** Akira Yoshida, Shunya Nagata, Yuka Okazaki, Hironari Hanaoka, Takahisa Gono, Masataka Kuwana

**Affiliations:** 1Department of Allergy and Rheumatology, Nippon Medical School Graduate School of Medicine, Tokyo 113-8602, Japan; y-okazaki@nms.ac.jp (Y.O.); t-gono@nms.ac.jp (T.G.); 2School of Medicine, Nippon Medical School, Tokyo 113-8602, Japan; 18shunya.n.00@nms.ac.jp; 3Division of Rheumatology, Department of Internal Medicine, Keio University School of Medicine, Tokyo 160-8582, Japan; hhanaoka@keio.jp; 4Scleroderma/Myositis Center of Excellence, Nippon Medical School Hospital, Tokyo 113-8603, Japan

**Keywords:** myositis, myositis-specific autoantibodies, anti-synthetase syndrome, anti-MDA5 antibodies, anti-Ro52 antibodies, interstitial lung disease

## Abstract

We aimed to dissociate the autoantibody response against the Ro52 protein in patients with anti-synthetase or anti-melanoma differentiation-associated gene 5 (MDA5) antibodies to explore the potential roles of different anti-Ro52 autoantibody responses in disease subclassification. This study used a single-center, prospective myositis cohort involving 122 consecutive patients with anti-synthetase antibodies identified by RNA immunoprecipitation (RNA-IP) and 34 patients with anti-MDA5 antibodies detected using enzyme immunoassay (EIA). Anti-Ro52 antibodies were measured using commercial EIA kits, while anti-Ro/SSA antibodies were identified using RNA-IP. Clinical features and outcomes were stratified according to two different patterns of autoantibody responses against Ro52, including “isolated anti-Ro52”, defined by positive anti-Ro52 and negative anti-Ro/SSA antibodies, and “anti-SSA-Ro52”, defined by positive anti-Ro52 and anti-Ro/SSA antibodies. Isolated anti-Ro52 positivity was the most prevalent autoantibody response in patients with both anti-synthetase (40/122; 32.8%) and anti-MDA5 antibodies (8/34; 23.5%). Isolated anti-Ro52 or anti-SSA-Ro52 positivity was associated with Gottron’s sign in patients with anti-synthetase antibodies, while in patients with anti-MDA5 antibodies, isolated anti-Ro52 positivity was associated with respiratory insufficiency at initial presentation and poor overall survival. Isolated anti-Ro52 positivity could be a potential biomarker for patient stratification; however, the clinical significance of dissociating isolated anti-Ro52 positivity from overall anti-Ro52 positivity was not evident.

## 1. Introduction

Idiopathic inflammatory myopathies (IIMs) are rare, systemic autoimmune rheumatic diseases (SARDs) arising from unknown causes. Muscle weakness due to skeletal muscle inflammation is the cardinal symptom of IIMs, while extramuscular organs such as skin, joints, lungs, heart, and gastrointestinal tracts are also affected [1]. IIMs are currently considered a highly heterogeneous disease spectrum presenting with different combinations of clinical manifestations, including myositis, distinctive skin rashes, interstitial lung disease (ILD), myocarditis, and arthritis [2].

A variety of circulating autoantibodies have been identified in patients with IIMs. Of these, myositis-specific autoantibodies (MSAs) are mutually exclusive and extremely useful for disease subclassification, treatment decisions, and prognostication [3]. Two major MSAs associated with ILD are anti-aminoacyl tRNA synthetase antibodies (anti-synthetase antibodies) and anti-melanoma differentiation-associated gene 5 (MDA5) antibodies [4]. Patients with anti-synthetase antibodies formulate a distinct subtype presenting with myositis, ILD, arthritis, Raynaud’s phenomenon (RP), unexplained fever, and mechanic’s hands, and a clinical entity called anti-synthetase syndrome (ASSD) has been proposed [5]. ILD in patients with anti-synthetase antibodies generally responds to immunosuppressive treatments; however, it often recurs with glucocorticoid tapering [6], and some patients exhibit progressive fibrosing ILD [7]. In contrast, anti-MDA5 antibodies are detected in patients with dermatomyositis (DM) or amyopathic dermatomyositis (ADM), who are often complicated with rapidly progressive ILD (RP-ILD), which is associated with early mortality [8]. However, clinical presentation, disease course, and prognosis are heterogeneous in patients with anti-synthetase or anti-MDA5 antibodies, and there is a significant need for further fine subclassification.

Autoantibodies against Ro/SSA particles were first identified in sera from patients with Sjögren’s syndrome [9] and later detected in sera from patients with other SARDs, such as systemic lupus erythematosus (SLE), systemic sclerosis (SSc), IIMs, and even undifferentiated connective tissue disease [10]. The Ro/SSA antigen is a cytoplasmic ribonucleoprotein complex comprising noncoding Y-RNAs and two protein components with molecular weights of 52 kDa (Ro52) and 60 kDa (Ro60), the latter two being the major targets of autoantibodies [11]. Importantly, Ro52 is also present in the nucleus in the form of an isolated protein uncoupled with Y-RNA, which could also be a target of autoantibodies (Figure 1) [12]. Autoantibodies against Ro52 are detected in patients with anti-synthetase antibodies or anti-MDA5 antibodies [13]. Previous studies reported that anti-Ro52 antibody positivity was associated with a symptomatic form of ILD [14], acute-onset ILD resistant to immunosuppressive treatments [15], and furthermore, RP-ILD and poor prognosis [14,16] in patients with anti-synthetase antibodies; however, the association between the presence of anti-Ro52 antibodies and unfavorable outcomes was not replicated in some studies [17,18], and controversy remains regarding the significance of anti-Ro52 antibody positivity in patients with anti-synthetase antibodies. On the other hand, in patients with anti-MDA5 antibodies, recent studies demonstrated a significant association between anti-Ro52 antibody positivity and unfavorable outcomes, such as developing RP-ILD and poor prognosis [13,19,20,21,22,23,24].

Originally, autoantibodies against Ro/SSA-related antigens were identified by double immunodiffusion and later by RNA-immunoprecipitation (RNA-IP) assay; currently, enzyme immunoassay (EIA) and line immunoassay (LIA) are widely available and can detect these antibodies conveniently in daily clinical practice. The majority of recent studies employed only EIA or LIA to detect anti-Ro52 antibodies in patients with IIMs, with which these studies failed to discriminate autoantibodies against Ro52 coupled with Y-RNAs and those against uncoupled Ro52. In this context, the present study aimed to dissociate the autoantibody response against Ro52, employing RNA-IP in combination with EIA for the detection of anti-Ro52 antibodies in patients with anti-synthetase antibodies or anti-MDA5 antibodies. Specifically, we investigated the prevalence of “isolated anti-Ro52 positivity” and “anti-SSA-Ro52 positivity” and their potential significance for disease subclassification and prognostication.

## 2. Materials and Methods

### 2.1. Patients

In this study, subjects were selected from a single-center, prospective cohort of patients with myositis. Consecutive adult patients (1) who visited the Scleroderma/Myositis Center of Excellence (SMCE), Nippon Medical School Hospital from August 2014 to March 2022 and in whom anti-synthetase antibodies were identified using RNA-IP or (2) who visited Keio University Hospital from April 2008 to July 2014 or SMCE from August 2014 to March 2022 and in whom anti-MDA5 antibodies were detected using EIA were enrolled, regardless of clinical diagnosis. The present study obtained approval from the institutional review board of Nippon Medical School Hospital (B-2020-127) and was conducted according to the tenets of the Declaration of Helsinki.

### 2.2. Autoantibody Profiling

Serum samples obtained from each patient at the initial visit and stored at −20 °C were subjected to comprehensive autoantibody profiling [25]. Anti-synthetase antibodies were identified with RNA-IP using cultured HeLa cell extracts, as described previously [26]. Silver staining was used to detect immunoprecipitated RNA components. The antigenic specificity of individual anti-synthetase antibodies was determined based on comparison with the precipitated RNA components using the prototype sera positive for anti-Jo-1, anti-PL-7, anti-PL-12, anti-EJ, anti-OJ, or anti-KS antibodies. Anti-MDA5 antibodies were detected using an in-house EIA [27] or a commercially available EIA kit (MESACUP^TM^, Medical and Biological Laboratories, Tokyo, Japan) [28].

Anti-Ro52 and anti-Ro60 antibodies were identified using commercial EIA kits (ORG652 and ORG660, ORGENTEC, Mainz, Germany) according to the manufacturer’s instructions. Briefly, 96-well microplates coated with purified Ro52 or Ro60 protein were incubated with patients’ sera diluted 1:100. Peroxidase-conjugated anti-human IgG and tetramethylbenzidine were used as the secondary antibody and the substrate for visualization, respectively. The antibody levels (0–200 U/mL) were calculated from the optical density at 450 nm with reference to a standard curve constructed using serially titrated calibrators. The cutoff value for both antibodies was determined to be 25.0 U/mL according to the datasheet provided by the manufacturer. Anti-Ro/SSA antibodies were judged positive when the serum sample immunoprecipitated Y-RNAs in RNA-IP.

### 2.3. Clinical Data

The following clinical data were obtained from the SMCE myositis database: age at diagnosis, sex, clinical diagnosis, arthritis (not due to osteoarthritis, crystal-induced arthropathy, or other noninflammatory causes), muscle weakness of proximal extremities or neck [29], ILD [30], mechanic’s hands, skin rashes specific to DM (heliotrope rash, Gottron’s sign/papules), palmar papules, skin ulcer [31], unexplained fever, and RP. Classification of PM/DM was based on the 2017 European League Against Rheumatism (EULAR)/American College of Rheumatology (ACR) classification criteria for IIMs [29], while the classification of SSc was based on the 2013 ACR/EULAR classification criteria [32]. We defined those fulfilling both PM/DM and SSc classification criteria as PM/DM-SSc overlap. The proposed criteria for interstitial pneumonia with autoimmune features (IPAF) by the European Respiratory Society/American Thoracic Society [33] were applied to patients who were not classified by the criteria listed above.

The onset of ILD was categorized as acute (<1 month), subacute (1–3 months), chronic (>3 months), and asymptomatic. Rapidly progressive ILD (RP-ILD) was defined as ILD with all the following three within one month from the onset of respiratory symptoms: (1) progressive dyspnea, (2) progressive hypoxemia, and (3) a worsening on chest radiograph or high-resolution CT (HRCT) [34]. Serum biomarkers, including creatine kinase (CK), C-reactive protein (CRP), ferritin, Krebs von den Lungen-6 (KL-6), and surfactant protein D (SP-D), at the initial presentation were recorded. Peripheral capillary oxygen saturation (SpO_2_), arterial blood gas analysis results, pulmonary function tests including percent predicted forced vital capacity (%FVC) and diffusing capacity for carbon monoxide (%DL_CO_), and chest HRCT were also noted. HRCT findings of ILD were classified into four categories [30,35]: (1) usual interstitial pneumonia (UIP), (2) nonspecific interstitial pneumonia (NSIP) and/or organizing pneumonia (OP), (3) diffuse alveolar damage (DAD), and (4) others. Malignancy that was diagnosed within five years before and after the disease diagnosis was documented. Initial treatment regimens, including the use of pulse methylprednisolone and immunosuppressive drugs, were also extracted from the database.

### 2.4. Outcome Measures

Overall survival (OS) and progression-free survival (PFS) were adopted as outcome measures. PFS was defined as the time from the diagnosis to the first progression, death, or the most recent visit. We captured disease progression as the occurrence of two or more of the following [6]: (1) deterioration of respiratory symptoms or clinical signs related to IIMs, (2) increase in ILD-related parenchymal abnormality on chest HRCT or radiograph, (3) ≥10% decrease in %FVC, ≥15% decrease in %DL_CO_, ≥10 Torr decrease in partial pressure of oxygen (PaO_2_), or ≥4% decrease in SpO_2_, and (4) emergence of clinical signs related to IIMs. The time of the last observation was 30 November 2022.

### 2.5. Statistical Analysis

First, we summarized the prevalence of anti-Ro52, anti-Ro60, and anti-Ro/SSA antibodies and assessed the diversity of autoantibody responses against Ro/SSA antigens in patients with anti-synthetase antibodies or anti-MDA5 antibodies. Specifically, we investigated the prevalence of “isolated anti-Ro52”, defined by positive anti-Ro52 and negative anti-Ro/SSA antibodies, and “anti-SSA-Ro52”, defined by positive anti-Ro52 and anti-Ro/SSA antibodies. Then, clinical features and outcomes were stratified according to isolated anti-Ro52 or anti-SSA-Ro52 positivity.

Continuous variables are presented as the median with the interquartile range (IQR), and comparisons were made using the Kruskal–Wallis test or Mann–Whitney U test, as appropriate. Fisher’s exact test was employed for the comparison of categorical variables, and *p* values were corrected by multiplying by the number of comparisons made. We did not perform statistical analysis for variables with more than 10% missing data. Survival analyses were performed using Kaplan–Meier curves, and the difference between groups was tested using the log-rank test. A two-sided *p* value < 0.05 was considered statistically significant. All statistical analyses were performed using R version 4.3.1 (R Foundation for Statistical Computing, Vienna, Austria).

## 3. Results

### 3.1. Patient Characteristics

From August 2014 to March 2022, we performed RNA-IP in 736 patients who visited SMCE with clinical suspicion of myositis or SSc. Of these, 122 patients were positive for anti-synthetase antibodies (Figure 2A). All 122 patients were included in the present study regardless of clinical diagnosis. The median age at diagnosis was 64 [IQR 53–71] years old, and 85 (69.7%) were female. The autoantigenic specificity of anti-synthetase antibodies was Jo-1 in 30 (24.6%), PL-7 in 21 (17.2%), PL-12 in 15 (12.3%), EJ in 32 (26.2%), OJ in 6 (4.9%), and KS in 20 (16.4%). Two patients were positive for two specificities of anti-synthetase antibodies concomitantly: one for both anti-Jo-1 and EJ and the other for both anti-EJ and KS antibodies. Clinical diagnoses included DM in 22 (18.0%), ADM in 20 (16.4%), PM/IMNM in 14 (11.5%), SSc in 4 (3.3%), PM/DM-SSc overlap in 8 (6.6%), and IPAF in 53 (43.4%). Treatment regimens and outcomes were evaluated in 108 patients in whom treatment was initiated in our institution.

Forty-seven consecutive patients with anti-MDA5 antibodies identified using EIA were enrolled. Of these, 34 patients were included in the analysis (Figure 2B). The median age at diagnosis was 52 [IQR 44–65] years old, and 22 (64.7%) were female. Clinical diagnosis was DM in 7 (20.6%) and ADM in 24 (70.6%). One patient was classified as PM/IMNM according to the 2017 ACR/EULAR criteria; however, several DM pathognomonic rashes, including the V sign and periungual erythema, were present. Another patient had neither skin rashes compatible with DM nor muscle weakness and was classified as IPAF. There was one unclassified patient who was intubated due to respiratory insufficiency at the initial visit, and manual muscle testing could not be evaluated appropriately.

### 3.2. Dissociation of Autoantibody Response against Ro/SSA Antigens

Of the 122 patients with anti-synthetase antibodies, 23 (18.9%) were positive for anti-Ro/SSA antibodies by RNA-IP, while 51 (41.8%) and 18 (14.8%) patients were positive for anti-Ro52 and anti-Ro60 antibodies by EIA, respectively (Figure 3A and Appendix A). Notably, 40/51 (78.4%) patients with anti-Ro52 antibodies by EIA were negative for anti-Ro/SSA antibodies by RNA-IP and therefore “isolated anti-Ro52” positive. Isolated anti-Ro52 positivity was the most prevalent pattern of autoantibody response against Ro/SSA antigens in patients with anti-synthetase antibodies, accounting for 40/122 (32.8%). Meanwhile, 11/51 (21.6%) patients with anti-Ro52 antibodies by EIA were also positive for anti-Ro/SSA by RNA-IP and therefore “anti-SSA-Ro52” positive.

For the 34 patients with anti-MDA5 antibodies, only two (5.9%) were positive for anti-Ro/SSA antibodies by RNA-IP, while anti-Ro52 and anti-Ro60 antibodies were detected with EIA in nine (26.5%) and one (2.9%), respectively (Figure 3B and Appendix A). Eight (88.9%) of the nine patients with anti-Ro52 antibodies by EIA were negative for anti-Ro/SSA antibodies by RNA-IP and therefore isolated as anti-Ro52 positive, which was, again, the most prevalent pattern of autoantibody response against Ro/SSA-related antigens (8/34; 23.5%). In contrast, only one (11.1%) of the nine patients with anti-Ro52 antibodies by EIA were positive for anti-Ro/SSA by RNA-IP (anti-SSA-Ro52 positive).

The prevalence of anti-Ro/SSA (18.9% vs. 5.9%, *p* = 0.109), anti-Ro52 (41.8% vs. 26.5%, *p* = 0.115), and anti-Ro60 antibodies (14.8% vs. 2.9%, *p* = 0.076) were all numerically higher in patients with anti-synthetase antibodies than in those with anti-MDA5 antibodies. There was no significant difference in the prevalence of isolated anti-Ro52 (32.8% vs. 23.5%, *p* = 0.401) or anti-SSA-Ro52 (9.0% vs. 2.9%, *p* = 0.465) positivity between patients with anti-synthetase and anti-MDA5 antibodies.

### 3.3. Clinical Characteristics and Outcomes Stratified by Isolated Anti-Ro52 or Anti-SSA-Ro52 Positivity

Next, we investigated the clinical significance of stratifying patients with anti-synthetase antibodies or anti-MDA5 antibodies according to isolated anti-Ro52 or anti-SSA-Ro52 positivity. Patients with anti-synthetase antibodies were divided into three groups: (i) isolated anti-Ro52 positive (*n* = 40; 32.8%), (ii) anti-SSA-Ro52 positive (*n* = 11; 9.0%), and (iii) anti-Ro52 negative (*n* = 71; 58.2%) (Table 1). No significant differences were observed in demographics and clinical diagnosis between the three groups. The distribution of antigenic specificity of anti-synthetase antibodies differed significantly (*p* = 0.005); anti-Jo-1 and anti-PL-12 antibodies were more prevalent in patients with isolated anti-Ro52 or anti-SSA-Ro52 positivity, while anti-PL-7, anti-OJ, and anti-KS antibodies were more prevalent in those without anti-Ro52 antibodies. Notably, isolated anti-Ro52 or anti-SSA-Ro52 positivity was associated with the presence of Gottron’s sign (60.0% vs. 54.5% vs. 35.2%, *p* = 0.032). No significant difference was noted in the prevalence of ILD, ILD onset, the frequency of RP-ILD, chest HRCT pattern, or serum biomarkers at initial diagnosis between the three groups. Importantly, there was no significant difference in clinical features between patients with isolated anti-Ro52 positivity and those with anti-SSA-Ro52 positivity. Treatment regimens were comparable between the three groups except for intravenous cyclophosphamide (IVCY), which was more likely to be used in those with anti-SSA-Ro52 positivity (18.8% vs. 71.4% vs. 23.1%, *p* = 0.022). For the outcome measures, there was no significant difference in OS (*p* = 0.444) (Figure 4A) or PFS (*p* = 0.833) (Figure 4B) between the three groups.

For the patients with anti-MDA5 antibodies, 33 (97.1%) were either isolated anti-Ro52 positive or anti-Ro52 negative, and only one patient (2.9%) presented with anti-SSA-Ro52 positivity; therefore, we compared clinical characteristics between patients with and without isolated anti-Ro52 positivity (Table 2). Patients with isolated anti-Ro52 positivity were older than those without (65 [61–68] years old vs. 48 [41–60] years old, median [IQR]; *p* = 0.025). Of note, RP-ILD was more prevalent (75.0% vs. 30.8%, *p* = 0.042), the partial pressure of oxygen (PaO_2_)/fraction of inspired oxygen (FiO_2_) ratio was lower (320.0 [270.2–373.4] vs. 416.7 [346.2–437.6], median [IQR]; *p* = 0.029), and the alveolar–arterial oxygen gradient (A-aDO_2_) was higher (41.9 [30.4–54.2] vs. 19.2 [10.1–33.6], *p* = 0.019) in patients with isolated anti-Ro52 positivity than in those without. Patients with isolated anti-Ro52 positivity had significantly worse OS than those without (Figure 4C, *p* = 0.002). We did not analyze PFS in patients with anti-MDA5 antibodies since there was a statistically significant difference in OS.

## 4. Discussion

The present study successfully dissociated the autoantibody response against the Ro52 protein in patients with anti-synthetase antibodies or anti-MDA5 antibodies using RNA-IP in combination with EIA. To the best of our knowledge, this is the first study that aimed to stratify patients according to two distinct specificities of anti-Ro52 antibodies. Notably, isolated anti-Ro52 positivity, defined as positive anti-Ro52 antibodies by EIA and negative anti-Ro/SSA by RNA-IP, was the most prevalent pattern of autoantibody response against Ro/SSA-related antigens both in patients with anti-synthetase antibodies or anti-MDA5 antibodies, which accounted for 32.8% and 23.5%, respectively. In patients with anti-synthetase antibodies, isolated anti-Ro52 or anti-SSA-Ro52 positivity was associated with Gottron’s sign; however, we failed to show the clinical significance of dissociating isolated anti-Ro52 and anti-SSA-Ro52, including outcome stratification. In contrast, in patients with anti-MDA5 antibodies, isolated anti-Ro52 positivity almost completely corresponded to anti-Ro52 positivity, which was associated with an increased frequency of RP-ILD and respiratory insufficiency at initial presentation, namely, a lower PaO_2_/FiO_2_ ratio and higher A-aDO2. Furthermore, isolated anti-Ro52 positivity was associated with poor OS in those with anti-MDA5 antibodies.

Based on a targeted literature review using PubMed, previous studies utilized EIA or LIA alone for detecting anti-Ro52 antibodies and reported the prevalence of anti-Ro52 antibodies as 21.4–65.9% in patients with anti-synthetase antibodies [13,14,15,16,17,18,36,37,38,39,40,41] (Appendix A) and 31.5–74.7% in those with anti-MDA5 antibodies [13,19,20,21,22,23,24] (Appendix A), respectively, which are compatible with the prevalence in the present study.

Anti-Ro/SSA antibodies were detected in 18.9% of patients with anti-synthetase antibodies with RNA-IP, consistent with the finding from a previous multicenter study that reported the prevalence of anti-Ro/SSA antibodies as 31/166 (18.7%) in patients with anti-synthetase antibodies using RNA and protein-IP [42]. In contrast, only 5.9% of patients with anti-MDA5 antibodies were positive for anti-Ro/SSA antibodies. Meanwhile, the prevalence of isolated anti-Ro52 positivity was relatively comparable between the two groups (32.8% vs. 23.5%), suggesting that different pathophysiology might play a role in the production of anti-Ro/SSA antibodies and autoantibodies targeting Ro52 uncoupled from Y-RNA. Patients with anti-synthetase antibodies are often complicated by other SARDs [43], implying that expansion and diversification of the autoantibody responses targeting multiple autoantigens precedes disease onset, as seen in rheumatoid arthritis or SLE [44]. In contrast, patients with anti-MDA5 antibodies rarely have overlapping SARDs and present with an acute disease course, and environmental factors such as viral infection are implicated in the pathogenesis [45].

Our results highlight the predominant isolated anti-Ro52 positivity in patients with anti-synthetase or anti-MDA5 antibodies. Recently, Shao et al. investigated 150 patients with anti-Ro52 antibodies without anti-Ro60 antibody positivity in their ILD cohort and found that 138 were positive for either anti-synthetase (*n* = 88; 61.5%) or anti-MDA5 (*n* = 42; 38.5%) [46]. Overall, these results imply that isolated anti-Ro52 positivity could be the most prevalent and unique pattern of autoantibody response against Ro/SSA-related antigens in patients with IIMs and ILD.

Either isolated anti-Ro52 or anti-SSA-Ro52 positivity was associated with the presence of Gottron’s sign in patients with anti-synthetase antibodies. This is in line with a recent study by Gui et al., in which the authors investigated 267 patients with IIMs and ILD (173 were positive for anti-synthetase antibodies) and reported the association between anti-Ro52 antibody positivity and Gottron’s sign [13]. The Ro52 protein has E3 ubiquitin ligase activity and is involved in the ubiquitination of cellular proteins, including interferon regulatory factors (IRF) [12]; consequently, isolated Ro52 negatively regulates type I interferon (IFN) production by targeting IRF3 and IRF7 [47,48]. Anti-Ro52 antibodies might amplify type I IFN signaling by blocking the regulatory activity of Ro52 proteins [49], which could be the underlying mechanism for the association between anti-Ro52 antibody positivity and Gottron’s sign. In patients with anti-MDA5 antibodies, isolated anti-Ro52 positivity was associated with RP-ILD and respiratory insufficiency at the initial visit and decreased OS. Our result is consistent with recent cohort studies reporting a significant association between anti-Ro52 antibody positivity by LIA and early death in patients with anti-MDA5 antibodies [13,20,23,24], supporting the hypothesis that anti-Ro52 antibodies could be a promising prognostic biomarker in these conditions.

We acknowledge the limitations of the present study. This is a retrospective study using a single-center cohort, and we are aware of the selection bias inherent to this type of study. Additionally, we failed to adjust for possible confounders, including older age, in the survival analyses due to the limited number of deceased cases, especially in patients with anti-MDA5 antibodies. Our findings should be further explored and verified in future multicenter, prospective cohort studies.

## 5. Conclusions

Our study highlights that isolated anti-Ro52 positivity was the most frequent pattern of autoantibody response against Ro/SSA-related antigens in patients with anti-synthetase or anti-MDA5 antibodies, which could be a potential biomarker for disease subclassification; however, the clinical significance of dissociating isolated anti-Ro52 positivity from overall anti-Ro52 positivity is not evident from the present study.

## Figures and Tables

**Figure 1 diagnostics-13-03621-f001:**
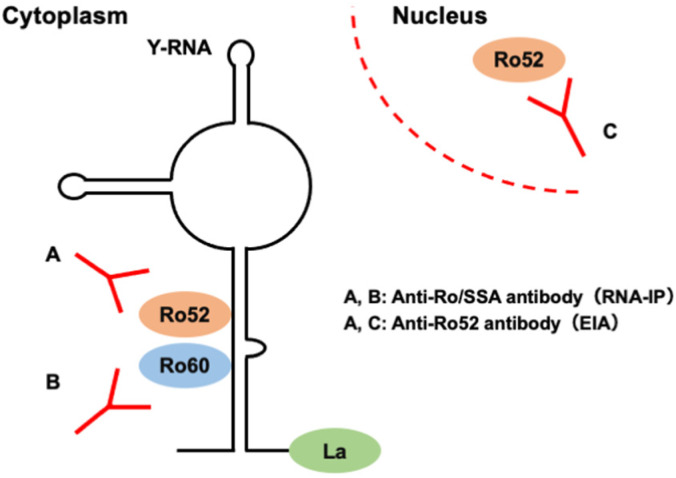
Structure of Ro/SSA antigen and the potential epitopes of anti-Ro/SSA or anti-Ro52 antibodies. RNA-IP identifies anti-Ro/SSA antibodies targeting Ro52 or Ro60 proteins coupled with Y-RNA (**A** and/or **B**), whereas EIA detects anti-Ro52 antibodies targeting Ro52 coupled or uncoupled with Y-RNA (**A** and/or **C**). EIA, enzyme immunoassay; RNA-IP, RNA immunoprecipitation.

**Figure 2 diagnostics-13-03621-f002:**
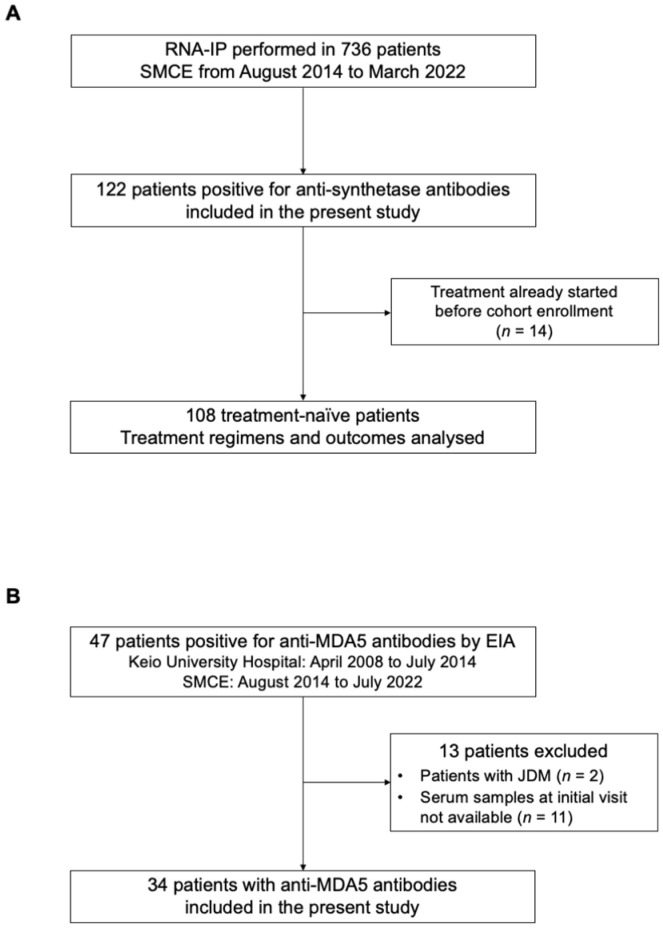
A flow chart of patient selection. (**A**) Patients with anti-synthetase antibodies, (**B**) patients with anti-MDA5 antibodies. EIA, enzyme immunoassay; JDM, juvenile dermatomyositis; MDA5, melanoma differentiation-associated gene 5; RNA-IP, RNA-immunoprecipitation; SMCE, Scleroderma and Myositis Center of Excellence.

**Figure 3 diagnostics-13-03621-f003:**
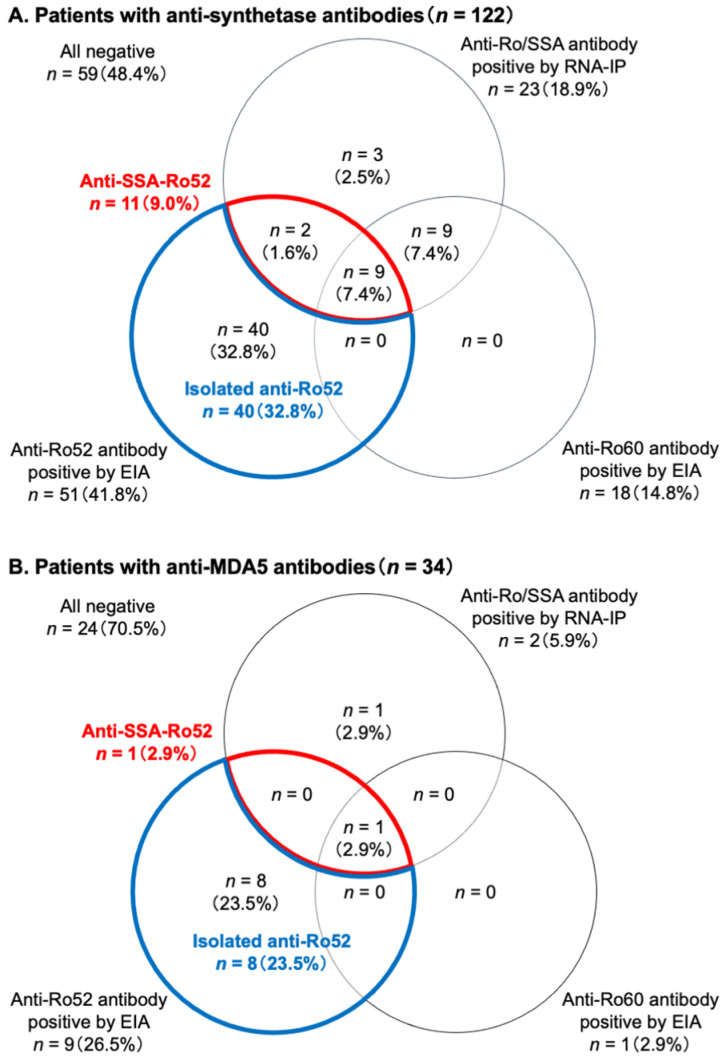
The prevalence of anti-Ro/SSA, anti-Ro52, and anti-Ro60 antibodies in patients with (**A**) anti-synthetase antibodies or (**B**) anti-MDA5 antibodies. EIA, enzyme immunoassay; MDA5, melanoma differentiation-associated gene 5; RNA-IP, RNA immunoprecipitation.

**Figure 4 diagnostics-13-03621-f004:**
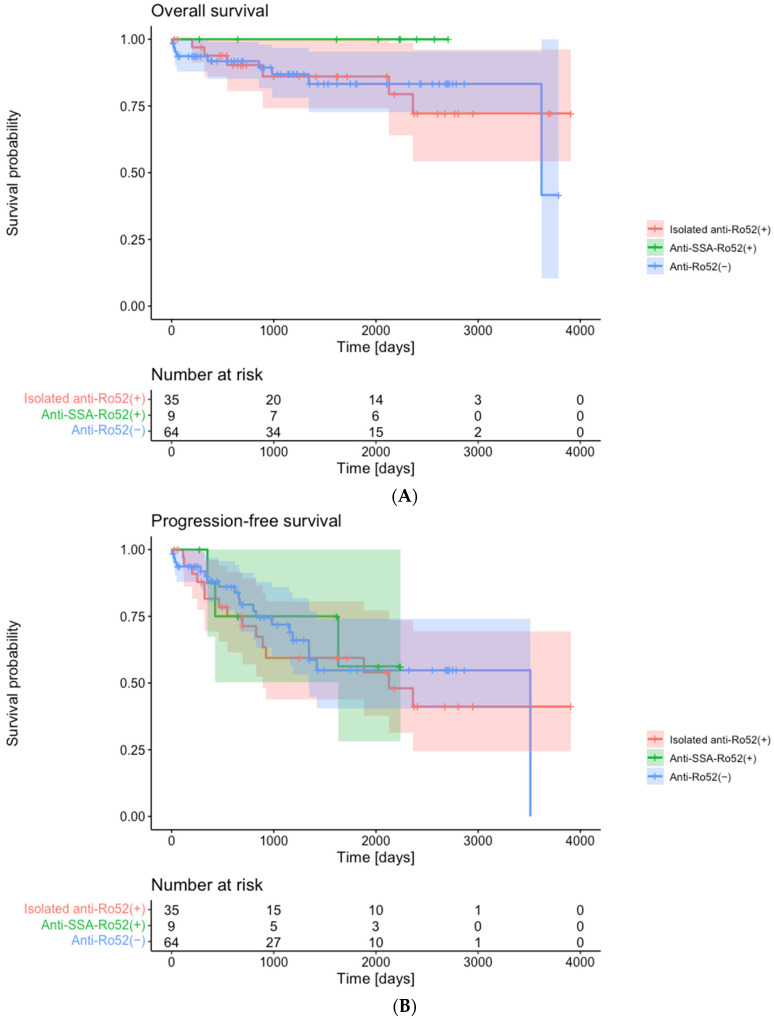
Outcomes stratified by the presence of different antibodies against Ro52. (**A**) Overall survival and (**B**) progression-free survival of patients with anti-synthetase antibodies stratified by isolated anti-Ro52 or anti-SSA-Ro52 positivity. (**C**) Overall survival of patients with anti-MDA5 antibodies stratified by isolated anti-Ro52 positivity. MDA5, melanoma differentiation-associated gene 5.

**Table 1 diagnostics-13-03621-t001:** Clinical characteristics of patients with anti-synthetase antibodies stratified by the presence of isolated anti-Ro52 and anti-SSA-Ro52 antibodies.

	Isolated Anti-Ro52 (+) *n* = 40	Anti-SSA-Ro52 (+) *n* = 11	Anti-Ro52 (−) *n* = 71	*p* *
Age at diagnosis (years)	64 [58–70]	60 [54–65]	65 [53–71]	0.356
Female	30 (75.0%)	9 (81.8%)	46 (64.8%)	0.379
Anti-synthetase antibodies				0.005
Anti-Jo-1	14 (35.0%)	5 (45.5%)	10 (14.1%)	
Anti-PL-7	2 (5.0%)	1 (9.1%)	18 (25.4%)	
Anti-PL-12	7 (17.5%)	3 (27.3%)	5 (7.0%)	
Anti-EJ	12 (32.0%)	1 (9.1%)	17 (23.9%)	
Anti-OJ	0	0	6 (8.5%)	
Anti-KS	4 (10.0%)	1 (9.1%)	14 (19.7%)	
Double positive	1 ** (2.5%)	0	1 *** (1.4%)	
Clinical diagnosis				0.814
DM	8 (20.0%)	3 (27.3%)	11 (15.5%)	
ADM	9 (22.5%)	2 (18.2%)	9 (12.7%)	
PM/IMNM	3 (7.5%)	2 (18.2%)	9 (12.7%)	
SSc	0	0	4 (5.6%)	
PM/DM-SSc overlap	3 (7.5%)	0	5 (7.0%)	
IPAF	17 (42.5%)	4 (36.4%)	32 (45.1%)	
Unclassified	0	0	1 (1.4%)	
DM-specific rash at diagnosis	20 (50.0%)	5 (45.5%)	23 (32.4%)	0.162
Heliotrope rash	7 (17.5%)	3 (27.3%)	9 (12.7%)	0.389
Gottron’s papules	5 (12.5%)	0	2 (2.8%)	0.133
Gottron’s sign	24 (60.0%)	6 (54.5%)	25 (35.2%)	0.032
Muscle weakness at diagnosis, *n* = 121	6 (15.0%)	4 (36.4%)	18 (25.7%)	0.233
Muscle weakness, *n* = 121	9 (22.5%)	4 (36.4%)	20 (28.6%)	0.582
ILD at diagnosis	39 (97.5%)	10 (90.9%)	65 (91.5%)	0.432
ILD	39 (97.5%)	10 (90.9%)	67 (94.4%)	0.423
ILD onset, *n* =120				0.812
No ILD	1 (2.6%)	1 (9.1%)	4 (5.6%)	
Acute	15 (39.5%)	4 (36.4%)	22 (31.0%)	
Subacute	6 (15.8%)	1 (9.1%)	11 (15.5%)	
Chronic	13 (34.2%)	5 (45.5%)	23 (32.4%)	
Asymptomatic or unclassified	3 (7.9%)	0	11 (15.5%)	
RP-ILD, *n* = 120	10 (26.3%)	2 (18.2%)	14 (19.7%)	0.708
HRCT pattern of ILD				0.796
No ILD	1 (2.5%)	1 (9.1%)	4 (5.6%)	
UIP pattern	3 (7.5%)	1 (9.1%)	4 (5.6%)	
NSIP and/or OP pattern	31 (77.5%)	8 (72.7%)	57 (80.3%)	
DAD pattern	3 (7.5%)	0	4 (5.6%)	
Others/undetermined	2 (5.0%)	1 (9.1%)	2 (2.8%)	
SpO_2_/FiO_2_ ratio, *n* = 103	452.4 [447.6–461.9]	461.9 [450.0–464.3]	457.1 [447.6–466.7]	NA
PaO_2_/FiO_2_ ratio, *n* = 80	375.2 [354.5–397.6]	450.7 [372.5–491.4]	417.1 [354.5–445.5]	NA
A-aDO_2_ (mmHg), *n* = 80	22.6 [13.71–31.6]	25.4 [9.1–34.2]	15.1 [6.5–27.5]	NA
%FVC, *n* = 88	74.6 [53.4–85.5]	78.1 [65.8–84.6]	83.6 [69.3–96.2]	NA
%DL_CO_, *n* = 80	59.8 [57.1–75.8]	71.3 [50.2–73.3]	66.9 [56.0–88.0]	NA
Arthritis	10 (25.0%)	5 (45.5%)	16 (22.5%)	0.260
Fever	7 (17.5%)	2 (18.2%)	16 (22.5%)	0.887
Mechanic’s hand	20 (50.0%)	5 (45.5%)	39 (54.9%)	0.823
Raynaud’s phenomenon, *n* =121	7 (17.5%)	4 (36.4%)	17 (24.3%)	0.369
Malignancy, *n* (%)	9 (22.5%)	1 (9.1%)	6 (8.5%)	0.098
CK (U/L), *n* = 119	92.5 [54.8–204.0]	161.5 [73.5–641.8]	129.0 [60.0–360.0]	0.313
CRP (mg/dL), *n* = 118	0.52 [0.07–2.37]	0.81 [0.14–1.40]	0.32 [0.08–1.21]	0.734
KL-6 (U/mL), *n* = 118	1091.0 [570.4–1573.1]	771.4 [480.7–1198.9]	947.2 [547.3–1683.6]	0.446
SP-D (ng/mL), *n* = 104	177.4 [112.2–308.3]	150.2 [123.2–243.8]	189.2 [119.3–404.3]	NA
Ferritin (ng/L), *n* = 70	229.0 [153.7–382.1]	163.0 [56.1–182.4]	142.1 [62.0–233.8]	NA
Any immunomodulatory treatment, *n* = 108	32 (91.4%)	7 (77.8%)	52 (81.2%)	0.358
Initial treatment regimen, *n* = 91				
Pulse methylprednisolone	15 (48.4%)	3 (42.9%)	24 (44.4%)	0.946
TAC	21 (65.6%)	4 (57.1%)	32 (61.5%)	0.894
CYA	3 (9.4%)	0	1 (1.9%)	0.277
MTX	1 (3.1%)	0	1 (1.9%)	1.000
AZA	0	0	2 (3.8%)	0.594
MMF	0	0	1 (1.9%)	1.000
IVCY	6 (18.8%)	5 (71.4%)	12 (23.1%)	0.022

A-aDO_2_, alveolar–arterial oxygen gradient; ADM, amyopathic dermatomyositis; AZA, azathioprine; CK, creatinine phosphokinase; CRP, C-reactive protein; CYA, cyclosporin A; DAD, diffuse alveolar damage; DL_CO_, diffusing capacity of lung for carbon monoxide; DM, dermatomyositis; FiO_2_, fraction of inspired oxygen; FVC, forced vital capacity; HRCT, high-resolution computed tomography; ILD, interstitial lung disease; IMNM, immune-mediated necrotizing myopathy; IPAF, interstitial pneumonia with autoimmune features; IVCY, intravenous cyclophosphamide; KL-6, Krebs von den Lungen-6; MTX, methotrexate; MMF, mycophenolate mofetil; NA, not applicable; NSIP, nonspecific interstitial pneumonia; OP, organizing pneumonia; PaO_2_, partial pressure of oxygen; PM, polymyositis; RP-ILD, rapidly progressive interstitial lung disease; SP-D, surfactant protein-D; SpO_2_, saturation of percutaneous oxygen; SSc, systemic sclerosis; TAC, tacrolimus; UIP, usual interstitial pneumonia; continuous and categorical variables are shown as the median [IQR] and *n* (%), respectively. * Comparison between the three groups. ** Positive for both anti-Jo-1 and anti-EJ antibodies. *** Positive for both anti-EJ and anti-KS antibodies.

**Table 2 diagnostics-13-03621-t002:** Clinical characteristics of patients with anti-MDA5 antibodies stratified by isolated anti-Ro52 antibody positivity.

	Isolated Anti-Ro52 (+) *n* = 8	Isolated Anti-Ro52 (−) *n* = 26	*p*
Age at diagnosis (years)	65 [61–68]	48 [41–60]	0.025
Female	7 (87.5%)	15 (57.7%)	0.210
Clinical diagnosis			0.285
DM	3 (38%)	4 (15%)	
ADM	4 (50%)	20 (77%)	
PM/IMNM	0	1 (3.8%)	
IPAF	0	1 (3.8%)	
Unclassified	1 (12.5%)	0	
Heliotrope rash	3 (37.5%)	5 (19.2%)	0.315
Gottron’s papule/sign	8 (100.0%)	23 (88.5%)	1.000
Palmar papules, *n* = 33	6 (75.0%)	15 (60.0%)	0.678
Skin ulcer	3 (37.5%)	2 (7.7%)	0.072
Muscle weakness, *n* = 33	3 (42.9%)	5 (19.2%)	0.320
ILD	8 (100.0%)	23 (88.5%)	1.000
RP-ILD	6 (75.0%)	8 (30.8%)	0.042
Arthritis	1 (12.5%)	14 (53.8%)	0.053
Fever	6 (75.0%)	12 (46.2%)	0.233
Malignancy	0	2 (7.7%)	1.000
SpO_2_/FiO_2_ ratio, *n* = 33	442.9 [395.6–454.8]	457.1 [451.2–466.7]	0.075
PaO_2_/FiO_2_ ratio, *n* = 33	320.0 [270.2–373.4]	416.7 [346.2–437.6]	0.029
A-aDO_2_ (mmHg), *n* = 33	41.9 [30.4–54.2]	19.2 [10.1–33.6]	0.019
CK (U/L)	199 [88–320]	136 [78–266]	0.855
CRP (mg/dL)	1.30 [0.21–2.89]	0.46 [0.20–1.27]	0.570
KL-6 (U/mL)	929.0 [762.6–1128.6]	657.8 [372.2–947.9]	0.084
SP-D (ng/mL) *n* = 33	107.8 [63.5–151.8]	79.0 [52.7–103.8]	0.355
Ferritin (ng/L), *n* = 33	967.1 [809.1–1521.0]	572.2 [221.3–1034.9]	0.093
Initial treatment regimen			
Pulse methylprednisolone	7 (87.5%)	22 (84.6%)	1.000
TAC	3 (37.5%)	17 (65.4%)	0.228
CYA	3 (37.5%)	6 (23.1%)	0.649
IVCY	7 (87.5%)	17 (65.4%)	0.385
Triple therapy (GC + TAC/CYA + IVCY)	6 (75.0%)	16 (61.5%)	0.681
JAK inhibitors	1 (12.5%)	0	0.235
Plasma exchange	1 (12.5%)	0	0.235

A-aDO2, alveolar–arterial oxygen gradient; ADM, amyopathic dermatomyositis; CK, creatinine phosphokinase; CRP, C-reactive protein; CYA, cyclosporin A; DAD, diffuse alveolar damage; DM, dermatomyositis; FiO_2_, fraction of inspired oxygen; GC, glucocorticoid; ILD, interstitial lung disease; IMNM, immune-mediated necrotizing myopathy; IPAF, interstitial pneumonia with autoimmune features; IVCY, intravenous cyclophosphamide; JAK, Janus kinase; KL-6, Krebs von den Lungen-6; PaO_2_, partial pressure of oxygen; PM, polymyositis; RP-ILD, rapidly progressive interstitial lung disease; SP-D, surfactant protein-D; SpO_2_, saturation of percutaneous oxygen; TAC, tacrolimus. Continuous and categorical variables are shown as the median [IQR] and *n* (%), respectively.

## Data Availability

The data presented in this study are available upon request from the corresponding author. The data are not publicly available due to privacy restrictions.

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
