# Peer review of "Dissociating Autoantibody Responses against Ro52 Antigen in Patients with Anti-Synthetase or Anti-MDA5 Antibodies"

_diagnostics, 2023, doi:10.3390/diagnostics13243621_

Round 1
Reviewer 1 Report
Comments and Suggestions for Authors
It is a great manuscript that reveals the clinical relevance of anti-Ro52 antibodies positivity in patients with anti-synthetase syndrome or anti-MDA5. The authors found a different clinical phenotype in anti-synthetase patients and poor outcome in anti-MDA5, which emphasize the relevance of the anti-Ro52 determination for a better assessment of the patient prognosis. Congratulations for your great work.
Author Response
We thank a favorable opinion on our manuscript.
Reviewer 2 Report
Comments and Suggestions for Authors
In the present study, the authors aimed to dissociate the autoantibody response against Ro52 protein in patients with anti-synthetase or anti-melanoma differentiation-associated gene 5 (MDA5) antibodies to explore the potential roles of different anti-Ro52 autoantibody responses in disease subclassification.
The authors present exciting findings, but there are some comments to improve the current study.
1. Please state the purpose of the article more clearly. Please emphasize the novel
aspect of the article.
2. What is the methodology of RNA-IP and EIA for autoantibody profiling in the study? Please mention it very briefly in the method and material section.
3. In the discussion section, I would suggest the authors add on literature review in support of their results.
Author Response
We appreciate insightful comments, which helped us improve the quality of our manuscript. Please see below for our point-by-point responses. Kindly note that the page number refers to the tracked version of the revised manuscript.
- Please state the purpose of the article more clearly. Please emphasize the novel aspect of the article.
Response:
According to this reviewer’s comment, we have updated the Introduction and Discussion parts as follows:
Third paragraph, Page 2 and first paragraph, Page 3:
In this context, the present study aimed to dissociate the autoantibody response against Ro52 employing RNA-IP in combination with EIA for the detection of anti-Ro52 antibodies in patients with anti-synthetase antibodies or anti-MDA5 antibodies. Specifically, we investigated the prevalence of “isolated anti-Ro52 positivity”, and “anti-SSA-Ro52 positivity”, and their potential significance for disease subclassification and prognostication.
First paragraph, Page 11:
The present study successfully dissociated the autoantibody response against Ro52 protein in patients with anti-synthetase antibodies or anti-MDA5 antibodies using RNA-IP in combination with EIA. To the best of our knowledge, this is the first study aiming to stratify the patients according to two distinct specificities of anti-Ro52 antibodies.
- What is the methodology of RNA-IP and EIA for autoantibody profiling in the study? Please mention it very briefly in the method and material section.
Response:
In line with this reviewer’s comment, we have added explanations in the Materials and Methods section as follows:
Third paragraph, Page 3:
Anti-synthetase antibodies were identified by RNA-IP using cultured HeLa cell extracts, as described previously [21]. Silver staining was used to detect immunoprecipitated RNA components.
Fourth paragraph, Page 3:
Anti-Ro52 and anti-Ro60 antibodies were identified using commercial EIA kits (ORG652 and ORG660, ORGENTEC, Mainz, Germany), according to the manufacturer’s instruction. Briefly, 96-well microplate coated with purified Ro52 or Ro60 protein was incubated with patient’s sera diluted 1:100. Peroxidase-conjugated anti-human IgG and tetramethylbenzidine were used as the secondary antibody and the substrate for visualization, respectively. The antibody levels (0–200 U/mL) were calculated from the optical density at 450 nm, with reference to a standard curve constructed using serially titrated calibrators.
- In the discussion section, I would suggest the authors add on literature review in support of their results.
Response:
We performed a targeted literature review on MEDLINE using PubMed to retrieve articles investigating the prevalence and clinical significance of anti-Ro52 antibodies in patients with anti-synthetase or anti-MDA5 antibodies. We have provided the summary of literature review as Supplementary Table S1 and S2. We also incorporated some additional references resulting from the literature review. The Discussion part has been updated as follows. Please note that the Reference list has also been updated.
Second paragraph, Page 11:
Based on a targeted literature review using PubMed, previous studies utilized EIA or LIA alone for detecting anti-Ro52 antibodies and reported the prevalence of anti-Ro52 antibodies as 21.4–65.9% in patients with anti-synthetase antibodies [13-18,36-41] (Supplementary Table S1) and 31.5–74.7% in those with anti-MDA5 antibodies [13,19-24] (Supplementary Table S2), respectively, which were compatible with the prevalence in the present study.